# In Situ Monitoring and Assessment of Ischemic Skin Flap by High-Frequency Ultrasound and Quantitative Parameters

**DOI:** 10.3390/s24020363

**Published:** 2024-01-07

**Authors:** Da-Ming Huang, Shyh-Hau Wang

**Affiliations:** 1Department of Computer Science and Information Engineering, National Cheng Kung University, Tainan 70101, Taiwan; sgogo0228@hotmail.com; 2Institute of Medical Informatics, National Cheng Kung University, Tainan 70101, Taiwan

**Keywords:** high-frequency ultrasound, quantitative parameter, WMC Nakagami imaging, skin flap, ischemic necrosis

## Abstract

Skin flap surgery is a critical procedure for treating severe skin injury in which post-surgery lesions must well monitored and cared for noninvasively. In the present study, attempts using high-frequency ultrasound imaging, quantitative parameters, and statistical analysis were made to extensively assess variations in the skin flap. Experiments were arranged by incising the dorsal skin of rats to create a skin flap using the chamber model. Measurements, including photographs, 30 MHz ultrasound B-mode images, skin thickness, echogenicity, Nakagami statistics, and histological analysis of post-surgery skin flap, were performed. Photograph results showed that color variations in different parts of the skin flap may readily correspond to ischemic states of local tissues. Compared to post-surgery skin flap on day 7, both integrated backscatter (*IB*) and Nakagami parameter (*m*) of the distal part of tissues were increased, and those of the skin thickness were decreased. Overall, relative skin thickness, *IB*, and *m* of the distal part of post-surgery skin flap varied from 100 to 67%, −66 to −61 dB, and 0.48 to 0.36, respectively. These results demonstrate that this modality and quantitative parameters can be feasibly applied for long-term and in situ assessment of skin flap tissues.

## 1. Introduction

Skin is the largest organ of the human body and provides many essential functions, such as isolation of environmental pathogens, resistance to external attacks, control of water, and thermoregulation [1]. Inevitably, the skin may suffer different injuries from various external attacks throughout one’s lifespan. Most minor skin injuries may recover by the natural wound healing process in conjunction with proper treatments. Extensive procedures, such as skin flap and graft surgery [2], are necessarily and commonly employed to treat severe skin injuries. The procedure of skin flap surgery is typically performed by lifting and incising a certain area of the skin where sufficient blood supply circulating between donor site and recipient site [3] is maintained, and which differs from that of the graft surgery without blood supply. The procedures of skin flap surgery are generally less complicated than those of major surgeries. However, patients with skin flap surgery might still undergo complications, such as total loss of microsurgically transferred flaps (1–5%), partial flap necrosis with free flaps (7–20%), or pedicled flaps (20–33%) [4]. The recipient tissues of the skin flap surgery might also experience ischemia/reperfusion (I/R) injury [5], which consequently leads to insufficient blood supply and subsequently ischemic necrosis. Corresponding to I/R injury, the resultant reactive oxygen species (ROS) and inflammatory cytokines may further accumulate and then damage the wound skin tissues by apoptosis or necrosis process [6]. Therefore, it is crucial to long-term and noninvasively assess the non-buried flap tissue from patients undergoing skin flap surgery to provide appropriate treatment and to avoid the development of possible complications. Typical diagnostic indicators for the assessment of the skin flap include temperature, swelling, and color of the tissue [7] measured by photograph-based modalities, such as skin photography and dermoscope. Nevertheless, in addition to the accuracy of measurements of these indicators being largely dependent on physician’s experience, the related applicable medical devices are difficult to use long-term and in the in situ assessment of the buried flap and underlying tissues [8]. These issues could certainly be better resolved by considering the use of medical imaging modalities.

To date, X-ray computerized tomography (CT), optical coherence tomography (OCT), reflectance confocal microscopy (RCM), multispectral optoacoustic tomography (MSOT), and ultrasound imaging are commonly available modalities for noninvasively imaging and diagnosing soft tissues. Despite having higher spatial resolution, CT is still not an adequate tool for continuously monitoring the conditions of skin flap due to ionizing radiation consideration. OCT and RCM are reportedly able to image the tissue at a depth of 1–2 mm [9] and 200–300 μm [10], respectively, and are not suitable to measure and diagnose thicker skins of cutaneous diseases [11,12]. Optoacoustic imaging in conjunction with contrast agents tends to be a novel noninvasive modality for imaging the skin flap, and more verifications and necessary approvals are needed for it to be adopted in clinical applications [12]. Moreover, clinical use of ultrasound scanners with a wave frequency of less than 10 MHz are a spatial resolution short of differentiating fine structure in tissues, and the resolution of ultrasound imaging may be further improved by an increase in wave frequency. It has been shown that 50 MHz high-frequency ultrasound (HFUS) imaging has a promising image resolution of up to 40 μm, capable of differentiating layers of tissue structures, including forearm, cheek, eye corner [13]. HFUS has also demonstrated the ability to diagnose inflammation responses [14,15] as considered for providing better treatment and care for an increase in skin flap survival as well as to characterize the skin tissue properties and severity of the cutaneous diseases in situ [16,17].

Despite the abovementioned advantages and capability for assessing skin lesions, the quality and interpretation of HFUS images are still affected by the settings of the utilized ultrasound system, the acoustic properties of interrogated tissues, and the operators’ experience [18,19]. To address these issues, quantitative ultrasound (QUS) parameters, such as integrated backscatter (*IB*) and statistical parameters, calculated and estimated from ultrasound radio-frequency (RF) signals, were developed. Ultrasound shear wave elasticity is also capable of quantitatively measuring tissue elastic properties [18], and this was not adopted for measurement in this study, owing to several considerations such as the possible bio-effect on the vulnerable skin flap associated with the higher acoustic energy generated from an additional push transducer as well as the time constraint of scanning the whole skin flap tissue, alongside image spatial resolution. *IB* corresponds to echogenicity and may be estimated from the acquired backscattering signals over the bandwidth of incident ultrasound frequency [20]. It may correlate *IB* to scatterer size and density of interrogated tissues for use in differentiating stable and unstable vascular plaques [21]. On the other hand, the resultant echoes are also largely affected by acoustical properties and the arrangement of scatterers in the interrogated medium [18,22]. Consequently, a statistical approach by estimating the parameters of statistical models, such as Rician, K, homodyned K, and generalized K distributions [23,24], becomes feasible for adoption in differentiating different scattering signals of random nature processes. Specifically, statistical parameters are estimated from the probability density function (PDF) of ultrasonic backscattered envelopes, and may correlate with the concentration and arrangement of scatterers. The assessment of scattering signals by statistical approaches has advantages over intensity-based images or *IB* parameters as it can sensitively detect the variations in lesion tissue locally, with higher computational complexity. Therefore, a more general statistic, named the Nakagami distribution [25,26], than those just mentioned, provides statistical distributions. It was proposed and has been demonstrated to require much less computational effort to effectively characterize the properties of biological tissues from skin [27], muscle [28], and liver tissues [29]. Moreover, the Nakagami parameter (*m*) estimated from the Nakagami distribution is a quantitative parameter with relatively simple computation that can correlate to the shape of backscattering signal distribution [25]. The *m* parameter has been shown to be able to sensitively assess the severity of skin burn in a rat skin model. In response to the burning of skin with a 100 °C burn plate from 0 to 20 s, the *m* parameter tends to decrease from 0.76 to 0.45, respectively [27]. It is also capable of differentiating gastrocnemius muscle with or without Duchenne muscular dystrophy [28], in which the *m* of stage III (*m* = 0.85) is much larger than that of the control (*m* = 0.5) group. Moreover, degrees of liver fibrosis may also be distinguished by the *m* parameter [29], and tended to increase from 0.55 to 0.83 corresponding to that of Metavir scores from 0 to 4, respectively. All of these previous studies demonstrate that the incorporation of HFUS imaging with QUS parameters may be suitable for better assessing pathological changes in skin flap tissues.

To address this issue, a 30 MHz ultrasound imaging system capable of acquiring three-dimensional HFUS signals was developed. An in vivo animal model of skin flap tissues for long-term and noninvasive assessment by HFUS imaging and QUS parameters were arranged. The acquired data included photographs, HFUS signals, and images of in situ skin flap tissues. Three-dimensional HFUS and Nakagami *m* images were reconstructed and formed. Measurement and QUS parameters, including skin thickness, *IB*, and *m*, were subsequently performed and estimated to be utilized to assess pathological changes in local skin tissues. Experiments on each animal model proceeded from the surgery to day 7 post-surgery. The results of each measurement were subsequently better correlated and discussed with that of the corresponding histological sections.

## 2. Materials and Methods

### 2.1. Establishment of Skin Flap Model

The skin flap model from a total of six male Sprague-Dawley rats (8 weeks old, 327 ± 29 g) was prepared for this study. Surgery on the dorsal skin of each rat was performed to achieve a three-sided full-thickness skin flap of an area of 36 mm × 72 mm. The procedures of skin flap surgery included anesthetizing the rat with a 2% isoflurane (Panion & BF Biotech Inc., Tao Yuan, Taiwan) and then placing a silicon sheet beneath the skin flap to prevent blood supply from underlying tissues. Therefore, it only allowed for the inflow of blood supply from the proximal part of the flap. Subsequently, the skin flap and silicone sheet were clamped by a stainless fixation apparatus (FA) so that the occurrence of microcirculation reestablishment in the fixed tissue was prevented. The protocol of skin flap was adopted from a previous study [30]. The detailed arrangement of FA is given in Figure 1. All of the skin flap surgeries and experiment protocols were approved by the Institutional Animal Care and Use Committee of National Cheng Kung University, Tainan, Taiwan (approval number: 108180).

### 2.2. Experiment Arrangement

Long-term and in situ measurements, including the acquisition of photographs and ultrasound signals of the whole skin flap, were performed for each rat post-surgery at day 1, 3, 5, and 7. Image formation and QUS parameters for better assessing the variations in skin flap were subsequently acquired and estimated. Raster scanning was applied to move the ultrasound transducer, covering the scanning from the distal to proximal part of interrogated the skin flap tissue, as shown in Figure 2a. This allowed for a scanning area of 8 mm and 30 mm in lateral (X direction) and elevational (Y direction), respectively, with a line spacing of 200 µm on the skin flap, to be achieved. In total, 150 ultrasound B-mode images with an image depth of 13.86 mm were acquired from each animal on each day of measurement. The schematic arrangement of HFUS imaging system and the corresponding pictures as, respectively, given in Figure 2b and Figure 3, were comprised of a 30 MHz single element transducer (NIH Ultrasonic Transducer Resource Center, USC, Los Angeles, LA, USA), pulser (5073PR; Olympus, Center Valley, PA, USA), motor controller (DMC-18x2; Galil Motion Control Inc., Rocklin, CA, USA), pre-amplifier (LN1000A; Amplifier Research, PA, USA), arbitrary function generator (AFG3022B; Tektronix, Billerica, MA, USA), and an analog-to-digital converter (PIC-5114; Signatec Inc., Poway, CA, USA) of 8-bit depth and 8 MB onboard memory sizes.

The animal was fixed and placed on a custom-made rotatable platform where the position could be flexibly adjusted. Ultrasound scanning proceeded by placing the transducer inside a polymethylpentene bag, which was filled with distilled deionized water, for coupling the ultrasound wave between the transducer and the skin flap tissue. The characteristics of the applied transducer, including pulse-echo response and detailed information, are, respectively, given in Figure 4 and Table 1. The acquired A-line RF signals were filtered with a 30-MHz bandpass filter (BIF-30, Mini-Circuits, Brooklyn, NY, USA) and then digitized with a 250 MHz sampling frequency. Furthermore, 2D and 3D ultrasound images, QUS parameters (including *IB* and *m* parameter), and window modulation compounding (WMC) Nakagami imaging were subsequently reconstructed and estimated. Signal/image processing and estimation of the QUS parameters were conducted using MATLAB software (Version R2023a, The MathWorks, Natick, MA, USA), of which the overall workflow is given in Figure 5.

### 2.3. QUS Parameter Estimation and Imaging

QUS parameters, including *IB* and the Nakagami *m* parameter, and parameter-based imaging were estimated from the acquired ultrasound backscattering signals in the skin flap. The intensity-based *IB* [31] and the statistics-based Nakagami *m* parameter [25] have been demonstrated to be capable of quantitatively assessing the variations in scattering properties in biological tissues. As is well known, ultrasound backscattered signals from tissues vary accordingly with the shape, size, concentration, density, and elasticity of scatterers in the interrogated medium [31]. *IB* was estimated from the acquired ultrasound signals of interrogated medium with respect to that of from a perfect reflector using the following equation [32]:(1)$$IB=\frac{1}{f_{2}-f_{1}}\times \frac{\sum_{f_{1}}^{f_{2}}{\left|S_{r}(f)\right|}^{2}}{\sum_{f_{1}}^{f_{2}}{\left|S_{ref}(f)\right|}^{2}},$$
where $f_{1}$ and $f_{2}$, respectively, denote the lower and upper bound of −6 dB bandwidth frequencies; $S_{r}(\cdot )$ and $S_{ref}(\cdot )$ represent the spectra of backscattered signals from the flap tissue and a stainless steel reflector, respectively.

Nakagami statistics [25] have shown to be a general model capable of covering other statistical distributions by adjusting the distribution shape with the Nakagami *m* parameter. The *m* parameter has also demonstrated the ability to correlate to scatterer concentration, density, and arrangement in the interrogated tissues, and was applied to assess variations in various soft tissues lesions [22]. The Nakagami distribution for representing the PDF of ultrasonic backscattered envelope (*R*) is given as follows [22,23]:(2)$$f\left(R\right)=\frac{2m^{2}R^{2m-1}}{\Gamma \left(m\right)\Omega^{m}}e^{-\frac{m}{\Omega }R^{2}}\times U\left(R\right),$$
where Γ(·) and U(·), respectively, refer to the gamma function and unit step function; Ω and *m* represent scaling parameter and Nakagami parameter, respectively, and can be estimated by the following equations.
*Ω* = *E*(*R*^2^),(3)
and
(4)$$m=\frac{{\left[E(R^{2})\right]}^{2}}{{E[R^{2}-E(R^{2})]}^{2}},$$
in which E(·) denotes the statistical mean. The Nakagami *m* parameter is the shape parameter of Nakagami distribution, in which values of *m* < 1, =1, and >1 are correlating to pre-Rayleigh, Rayleigh, and post-Rayleigh statistical distribution, respectively. Specifically, the backscattering envelope with pre-Rayleigh distribution was found to result from a small number of scatterers in tissues, and that of Rayleigh distribution was associated with a large number of random located scatterers. As tissue is composed of numerous scatterers with regular arrangement, the corresponding statistics tends to be post-Rayleigh distributed [25]. Consequently, this allows for the correlation of *IB* and statistical parameters to quantitatively assess tissue properties.

In the present study, *IB* and *m* are QUS parameters estimated from a region-of-interest of 2.36 mm × 0.59 mm area covering the skin flap of the proximal, middle, and distal parts. The QUS parameters of the post-surgery skin flap of different days were correlated to the progress of the tissue necrosis. Moreover, WMC Nakagami imaging [33] was implemented to better observe variations in the skin flap macroscopically. The WMC Nakagami imaging ($M_{com}$) was carried out by compounding the conventional Nakagami images ($M_{i}$) of different window sizes, given as
(5)$$M_{com}\left(x,y\right)=\frac{1}{N}\sum M_{i}(x,y)$$
where N represents the number of $M_{i}$ over the sliding window of *i* times pulse length. Five $M_{i}$ (*N* = 5) over the sliding windows with *i* of 8, 9, 10, 11, and 12 times the pulse length were empirically estimated to be capable of preserving both the image smoothness and spatial resolution of WMC Nakagami images for assessing the local scatterer properties in the tissue. The procedures detailing post-processing and parameters estimation are given in Figure 5. The results were analyzed by ANOVA test using SPSS software (Version 17.0, SPSS Inc., Chicago, IL, USA), in which *p* < 0.05 was considered to be significant. The trends of QUS parameters with respect to necrosis progress as a function of post-surgery time were also estimated.

### 2.4. Histological Analysis

After finishing the ultrasound measurements at the end of seven days of post-surgery, the rats were sacrificed by inhalation with an overdose of CO_2_. A part of skin flap tissues of each animal was immediately excised and histological sections were made. Typical processes to make histological sections included the fixation and dehydration of the tissues, respectively, by 4% formaldehyde (ScyTek Laboratories Inc., Logan, UT, USA) and soaking fixed tissues in gradient ethanol solutions (Leica Biosystems Richmond Inc., Richmond, IL, USA) from 75% to nearly 100%. Subsequently, each tissue sample was cleared with xylene (Leica Biosystems Richmond Inc., Richmond, IL, USA), embedded in paraffin, and cut into a 10-µm-thick slice. The histological sections were stained with hematoxylin and eosin (H&E) for assessing cellular structure and morphology [34].

## 3. Results

A series of typical photographs, and 2D and 3D ultrasound B-mode images of the skin flap, as given in Figure 6, revealed the variations in skin flap surface and internal tissues from post-surgery rats that lasted for seven days. The *Y*-axis of images indicates the direction of skin flap from the proximal to distal part (ranging from 0 to 30 mm), in which the tissue variation from mild to severe ischemia could observed. The photographs and B-mode images may generally readily discern the variations in tissues associated with necrosis progression. Specifically, the color of skin flap tissue near the proximal part retains a pink color throughout the whole period of time-course measurement, while that of color in the distal part of skin flap was varying from pink to dark-red or even black color. Red and black colors in the photograph readily correspond to the occurrence of inflammatory response and necrosis of the skin flap, respectively. Moreover, changes in morphological and acoustic properties associated with the ischemia of tissues may be sensitively detected by 2D and 3D ultrasound B-mode images. Furthermore, 2D transversal B-mode images (Figure 6(ei)) corresponding to severe necrosis tissues near the distal part of skin flap were thinner thickness and had higher scattering intensity than that of near the proximal part, as seen in Figure 6(eiii). The skin thickness variation as a function of distance may also be better visualized from the reconstructed 3D images and sagittal 2D images in the *Y*-axis (Figure 6(aiv–eiv). All of these demonstrate that the pathological conditions of skin flap may be diagnosed by measurements of skin thickness and scattering intensity by ultrasound B-mode images.

Figure 7 detailed the variations in skin flap of different parts related to that of the skin thickness, in which the relative tissue thickness was achieved by normalizing data to that of day 0 post-surgery. In comparison to the relative skin thickness of the skin flap at day 0, those of the proximal, middle, and distal parts tended to decrease to 88.25 ± 7.60%, 81.54 ± 5.06%, and 67.01 ± 14.19% at day 7 post-surgery, respectively. Accompanying the progression of ischemia at day 7, the thickness of the skin flaps of the distal part tended to decrease significantly. According to analysis by ANOVA test, the relative skin thickness demonstrate a linear decrease with the progression of necrosis in the middle (*p* = 0.041) and distal (*p* < 0.001) parts of the skin flap. Moreover, spatiotemporal mapping (Figure 8a) was implemented by interpolating post-surgery data corresponding to different distances and times for comprehensively visualizing the information on skin thickness variation. An abrupt decrease in skin thickness located at around 25–30 mm in the *Y*-axis at day 2–3 post-surgery was found, and it might correlate to that of the region of tissue without sufficient blood supply. The results also showed that the skin flap maintained 80% of skin thickness at the region between 0 and 17 cm of the proximal part throughout 7-day measurement, and that of thickness between 17 and 30 cm of the distal part tended to decrease to 80% on day 3 post-surgery. Due to the condition of blood supply, variations in relative skin thickness in terms of relative standard deviation (RSD), as given in Figure 8b, in the region of the distal part were larger than that of the proximal part. Overall, all of the RSDs were generally smaller than 5%, which indicates the feasibility of applying skin thickness by ultrasound measurement to assess the progression of skin flap necrosis.

Furthermore, *IB* and *m* are QUS parameters that have been adopted to better assess variations in the acoustic properties of the skin flap. Figure 9 shows the results of *IB* at different regions in the skin flap as a function of post-surgery time. In response of the variations in the tissue properties to the condition of blood supply, the corresponding *IB* tends to increase with the progress of necrosis severity in the skin flap. The variations in *IB* of tissues in the distal part tend to increase from −65.74 ± 2.61 dB to −60.66 ± 2.02 dB between day 0 and day 7 post-surgery with a linear increased tendency (*p* = 0.001). Meanwhile, those of *IB*s of tissues in the middle and proximal parts seem to not vary largely. In general, the thickness and echogenicity of tissues in the distal part tend to decrease and increase, respectively, in response to the progression of ischemic necrosis.

Subsequently, statistical analysis with the Nakagami parameter was estimated for assessing the properties of scatterers, such as concentration and arrangement, in the skin flap [22]. The resultant *m* parameters were further transformed into 2D and 3D WMC Nakagami images, such as Figure 10 and Figure 11, for better comprehension of spatial variations in the skin flap. On day 0 post-surgery, various *m* parameters corresponding different statistical distributions to depths of skin flap were found, and this may be readily observed in 2D Nakagami images. Specifically, the epidermis and dermis in the skin flap on day 0 are distributed as *m* < 0.3 and range from *m* = 0.1 to 0.7, respectively. As the progresses of ischemic necrosis on days 1 to 7 post-surgery, the skin flap tissue in the distal part of epidermis was decreased to the region that is difficult to discern from Nakagami image, and while the statistics in the dermis was shifting toward Rayleigh-distributed. More thorough information about the whole skin flap may be explored by 3D Nakagami images. Ultrasound envelopes of dermis with a higher *m* parameter (red regions) in the distal part of skin flap were progressively extended toward that of the middle part of approximately 15 mm on day 7 post-surgery and were also verified by photograph results in Figure 6. Figure 12 shows the results summarizing the variations in *m* parameters in the regions of proximal, middle, and distal parts as a function of post-surgery time, which tend to vary, respectively, from 0.48 ± 0.04 to 0.44 ± 0.07, 0.48 ± 0.06 to 0.36 ± 0.07, and 0.44 ± 0.05 to 0.27 ± 0.05 relative to that of post-surgery tissues from day 0 to 7, in which the *m* parameter linearly decreases with the post-surgery time in the distal part of the skin flap (*p <* 0.001).

According to ultrasound B-mode images in Figure 6, skin flap thickness in the proximal, middle, and distal parts has a tendency of decreasing dependently on the degree of tissue ischemia. This result is consistent with that of histological sections acquired from skin flap tissues excised on day 7 post-surgery, as given in Figure 13, where the thickness of skin flap of different parts is estimated to be 3.17, 2.64, and 2.01 mm accordingly. Dehydration and collagen denaturation in the dermis of skin flap in accordance with the progress of necrosis may lead to the higher density of collagen fibers and the shrinkage of the extracellular matrix (ECM) [17], as can be seen in Figure 13. Moreover, immune cells may be observed in the middle part of skin flap, as marked by the yellow arrows in Figure 13, and correspond to more inflammatory response than the tissues of other regions.

## 4. Discussion

It is crucial to monitor the status of skin flap lesion in situ and long-term to provide appropriate treatments for improving its survival rate. To meet this need, assessment by the incorporation of HFUS imaging and QUS parameters was explored and developed. HFUS imaging has been demonstrated to have a spatial resolution capable of discerning tissues of the epidermis, dermis, and hypodermis from the skin flap of post-surgery rats on day 0. The cells and components in the ECM can also be resolved by HFUS as ultrasonic scatterers [35], and may reveal physiological variations in ischemia necrosis. Due to the composition of mainly condensed and flatly arranged keratinocytes, the hyperdermis tends to respond with higher scattering and echogenicity. On the other hand, the dermis comprising ECM with loosely arranged collagen and various proteins tends to lead to relatively lower ultrasound scattering strengths [36]. The hypodermis is the deepest layer of the skin, and as a resulted has the weakest echogenicity, partially because of its composition, namely, its amount of adipose tissues and higher ultrasound attenuation. Corresponding to the prolonging of post-surgery after day 0, the lesion of the skin flap lacked sufficient blood supply, and so a different degree of ischemia and necrosis may progressively occur and be readily observed by the variations in the skin’s surface from photograph results and those of cross-sectional regions with 30 MHz HFUS B-mode images, as given in Figure 6. Moreover, the variations in post-surgery skin flap associated with different degrees of ischemia in the distal, middle, and proximal parts may be better assessed by that of the thickness with ultrasound measurements. The results from Figure 7 evidently demonstrate that tissue thickness in the distal part tends to decrease more largely than other parts of the skin flap, and that the spatial variations in thickness may be comprehended in Figure 8. The decrease in thickness leads different layers of post-surgery skin to be merely distinguishable, and also tends to increase the tissue density. The results of histological analysis in Figure 13 provide further evidence about the variations in thickness and echogenicity in different parts of the skin flap. Specifically, the more necrotic tissues and compact ECM observed in the histological sections lead to the increase in echogenicity and *IB* in Figure 6 and Figure 9, as well as the decrease in the thickness of skin flap as a function of post-surgery time. Moreover, a large number of immune cells, such as monocytes and polymorphonuclear leukocytes, facilitated the production of free radicals [37] and ROS [38] in response to chronic inflammation and partially explain the increased echogenicity and *IB* in the middle part of the skin flap shown in Figure 9b. The immune cells in the histological sections of the skin flap have been proven to increase the mortality of post-surgery skin flap and also suggest that the corresponding tissues of ischemic necrosis are results of insufficient blood supply and inflammation. Furthermore, necrotic tissue in the deeper dermis of the distal part of post-surgery skin flap at day 7 was found, which may correspond to the degenerative vacuoles as appeared in the histological section [39].

To alleviate the effects of systemic factors, such as excitation signal, time-gain compensation, and the dynamic range in measurement of echogenicity and *IB* [23,40], statistical analysis of ultrasound backscattered strengths is used for sensitively assessing the variations in skin flap tissues corresponding to cellular compositions and the inflammation of tissues in terms of scatterers concentration and arrangement [40,41]. Specifically, WMC Nakagami imaging allows for better visualization and assessment of local variations in tissue properties in the skin flap. Due to the implementation of the sliding window size, the WMC Nakagami images nevertheless have a lower spatial resolution than that of ultrasound B-mode images when differentiating layers of skin tissues [33]. At day 0 post-surgery, the PDF of backscattering signals in the regions of epidermis, dermis, and hypodermis tended to distribute a range covering from *m* = 0.1 to 0.7. Accompanying the increasing lack of sufficient blood supply and variations in tissue, the backscattering signals in the distal part of the skin flap after day 0 post-surgery were distributed toward to Rayleigh statistics, as seen in Figure 10. This might correspond to variations in tissue components such as collagen fibers, ECM, and immune cells in the skin flap associated with changes in scatterers size, density, concentration, compressibility, and arrangement [41,42]. Specifically, collagen properties in terms of collagen content and bundle size were found to positively correlate with echogenicity, whereas those of bundle spacing was negatively correlated [43]. In accordance with the extent of ischemic lesion in the post-surgery skin flap, it caused a greater inflammatory response associated with immune cells, as shown in Figure 13b(i,ii), and a reduction in collagen fibers spacing, as shown in Figure 13b(iii). Consequently, the skin flap tissue of severe ischemic necrosis results in a higher backscattered strength and regular arrangement that leads the corresponding backscattering signals to have higher *m* parameters, as presented in WMC Nakagami images [44]. The results of histological sections may be correlated to that of QUS parameters, in which according to the results of ANOVA tests, *IB* and *m* parameters linearly increased and decreased to the progress of ischemic necrosis in the skin flap.

Both ultrasound B-mode and Nakagami images are able to visualize and assess local variations in lesion in the skin flap in situ and long-term. In particular, 3D B-mode and Nakagami images, as in those in Figure 6 and Figure 11, provide a more comprehensive means capable of assessing and diagnosing tissues from different views. Due to its utilization with HFUS, ultrasound B-mode imaging tends to have higher spatial resolution than that of WMC Nakagmai images when discerning variations in different structures in the skin flap, and the accompanying attenuation nevertheless degrades and complicates the image quality in the deeper region of tissues. On the other hand, statistics-based imaging provides a more quantitative means and tolerance to attenuation effects when differentiating variations in local tissues associated with the progress of post-surgery skin flap. However, due to the need to implement a certain size necessary for windowing the estimation *m* parameter, the WMC Nakagami image have less spatial resolution than that of the B-mode image. Moreover, inaccurate estimation of the *m* parameter corresponds to sub-resolvable effect on Nakagami imaging has been found in the studies of skin [27] and eyes [45] and needs to be taken into consideration. In the present study, the sub-resolvable effect on WMC Nakagami images was found to be in the skin flap tissues, with changes in their skin thickness owing to the progressively varying ischemic necrosis. Due to further dehydration and shrinkage of the ECM in the necrotic tissue of post-surgery skin flaps on day 7, the corresponding decrease and increase in skin thickness and scatterer density, respectively, tended to the increase echogenicity of the B-mode images and to degrade the WMC Nakagami images. The sub-resolvable effect can also be observed in the region of tissue interface between epidermis and dermis. This is due to the corresponding sliding window of signals for WMC Nakagami imaging resulting from the combination of both the epidermis (hyperechoic) and dermis (hypoechoic) [45]. Aside from the factors just discussed, the combination of HFUS image, QUS parameters, and WMC Nakagami images from 2D and 3D views provide a feasible means to comprehensively assess variations in skin flap tissues for consideration in related treatments.

## 5. Conclusions

In the present study, variations in skin flaps were extensively assessed by 30 MHz high frequency images, corresponding QUS parameters, and statistics-based imaging. The animal skin flap models were arranged, which allowed for long-term and noninvasive assessment in different regions of post-surgery skin flap associated with different degrees of ischemic necrosis. The 2D and 3D HFUS imaging provided sufficient spatial resolution and measurements of echogenicity and thickness for the in situ diagnosis of variations in local skin flap tissue. However, a challenge occurred in the differentiation of post-surgery tissue by the B-mode images due to the increasing attenuation and variation in echogenicity as a result of the decreasing thickness and increasing component density associated with severer necrosis in the skin flap. The statistical analysis and assessment by Nakagami parameters and WMC Nakagami imaging on the other hand are able to tolerate the attenuation effect, and this was correlated to the varying concentration and arrangement of related cells in the lesion tissue of post-surgery skin flaps. Nakagami *m* parameters and imaging did reveal a tendency for pre-Rayleigh backscattered signals, with shifting toward the Rayleigh distribution in the ischemic necrosis. The results of the histological sections disclose the corresponding cellular composition and arrangement information about different parts and depths of post-surgery skin flap. In general, the skin thickness, *IB*, and *m* of the distal part of skin flap varying more extensively than those of the other parts, in which the relative variations specifically corresponding to the post-surgery skin flap of day 0 and 7 range from 100% to 67%, −66 dB to −61 dB, and 0.48 to 0.36, respectively. The histological sections further verify the results of the primary factors of collagen shrinkage and denaturation. Despite their capability of tolerating attenuation, the WMC Nakagami images have lower spatial resolution than that of HFUS B-mode images and are accompanied by the sub-resolvable effect resulting from a sliding window of signals that cover different statistical distributions of tissues. Consequently, better long-term and in situ assessment of skin flap tissues may be achieved by combining HFUS image, QUS parameters, and WMC Nakagami images incorporated with 3D imaging capability. It certainly is worth further proceeding with clinical study as well as extensively exploring the optimal sliding window size for improving WMC Nakagami imaging and alleviating the sub-resolvable effect.

## Figures and Tables

**Figure 1 sensors-24-00363-f001:**
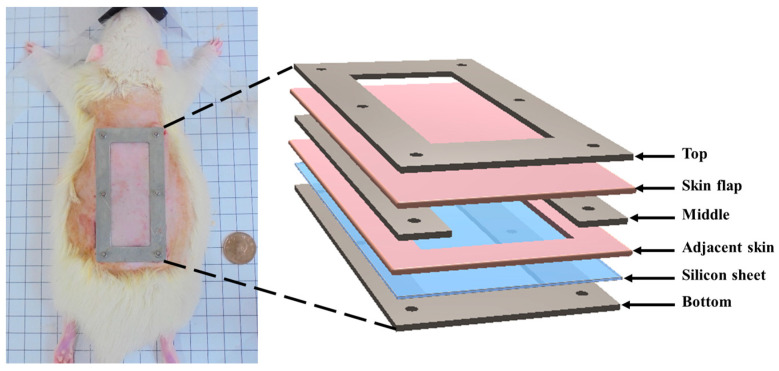
Components and application of fixation apparatus (FA) to the skip flap of a rat.

**Figure 2 sensors-24-00363-f002:**
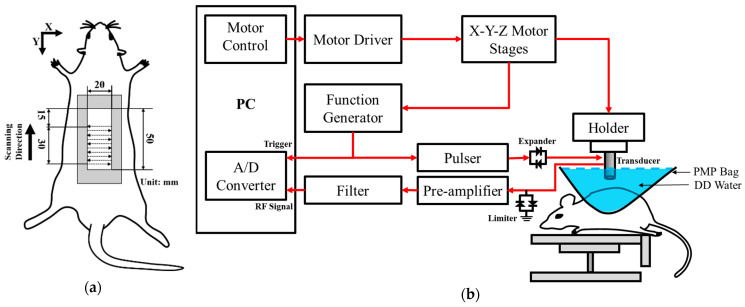
(**a**) Ultrasound scanning procedure; (**b**) schematic diagram of the high-frequency ultrasound system. A/D: analog-to-digital converter; DD: distilled deionized; PMP: polymethylpentene.

**Figure 3 sensors-24-00363-f003:**
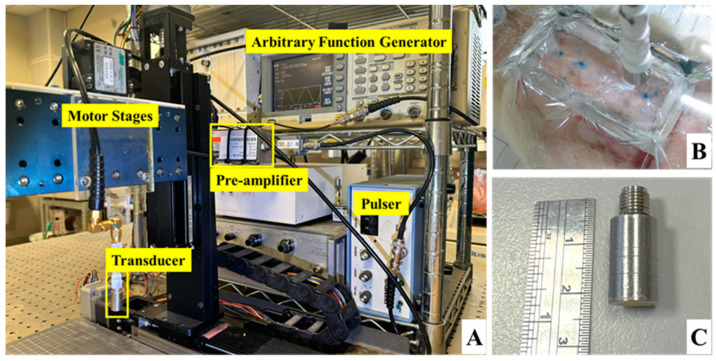
Pictures of (**A**) ultrasound imaging system, (**B**) arrangement of skin flap scanning, and (**C**) the applied 30 MHz transducer.

**Figure 4 sensors-24-00363-f004:**
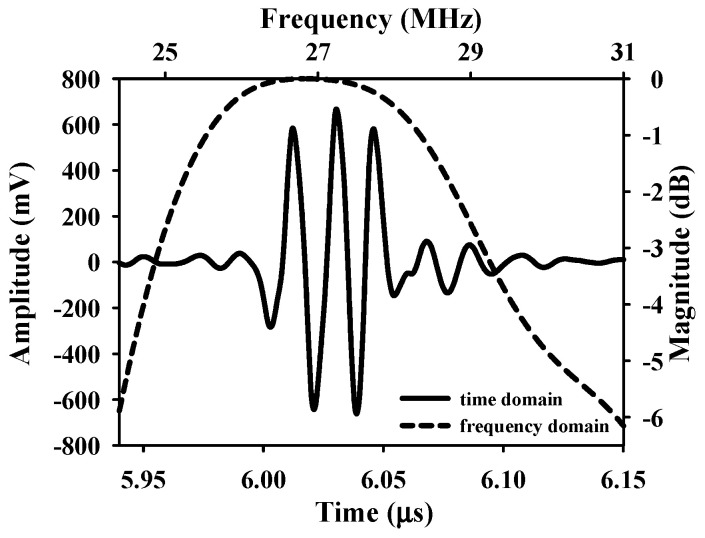
Pulse–echo response of the applied 30 MHz transducer.

**Figure 5 sensors-24-00363-f005:**
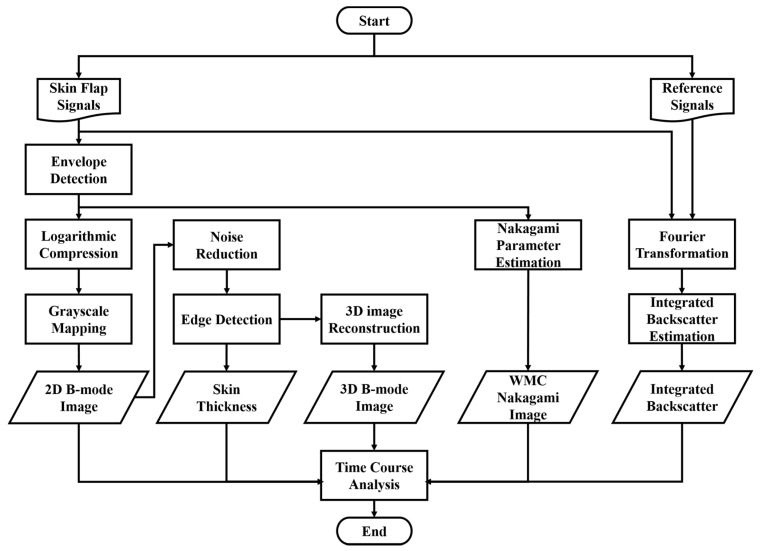
Procedures of ultrasound signal processing, imaging, and parameters estimations.

**Figure 6 sensors-24-00363-f006:**
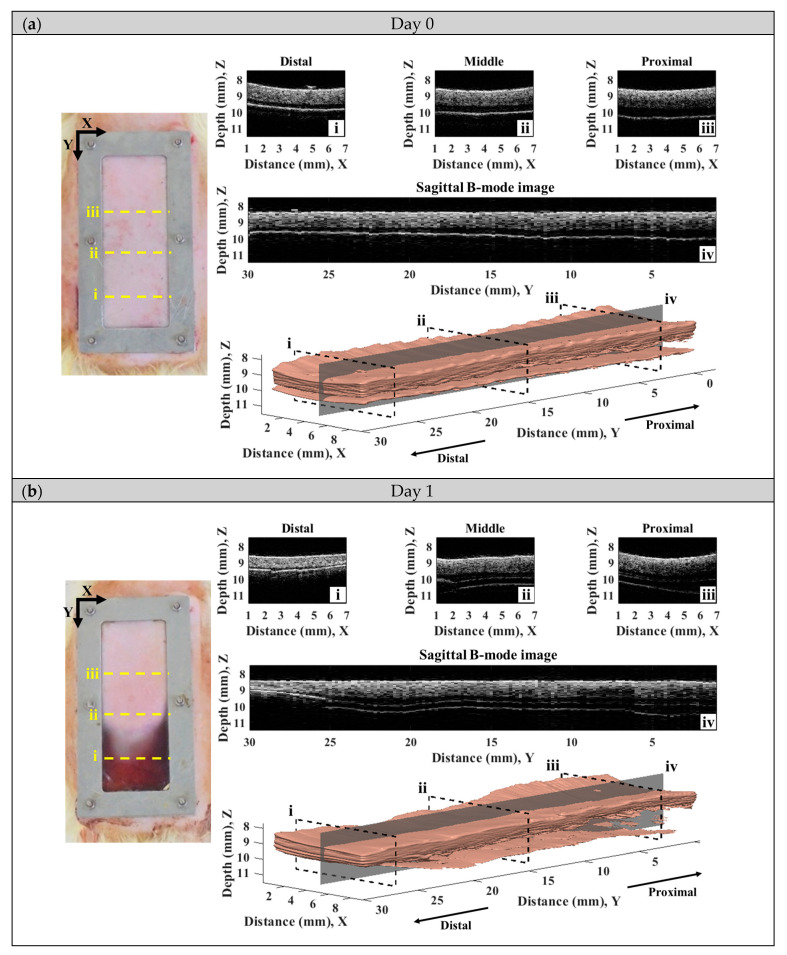
A series of typical photographs and 2D and 3D ultrasound B-mode images corresponding to day (**a**) 0, (**b**) 1, (**c**) 3, (**d**) 5, and (**e**) 7 post-surgery skin flap, in which 2D B-mode images of (**i**–**iii**) correspond to the transversal plane and that of (**iv**) to the sagittal plane.

**Figure 7 sensors-24-00363-f007:**
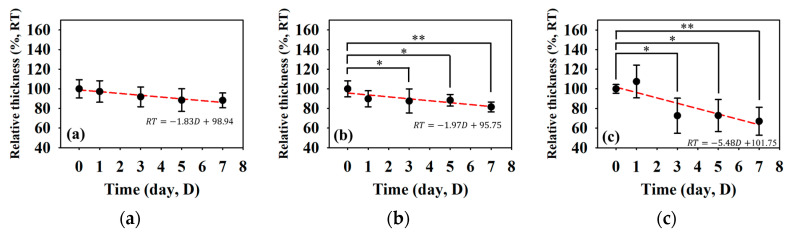
Relative skin thickness of the (**a**) proximal, (**b**) middle, and (**c**) distal parts of skin flap as a function of post-surgery time. (*N* = 6, *: *p* < 0.05, **: *p* < 0.005).

**Figure 8 sensors-24-00363-f008:**
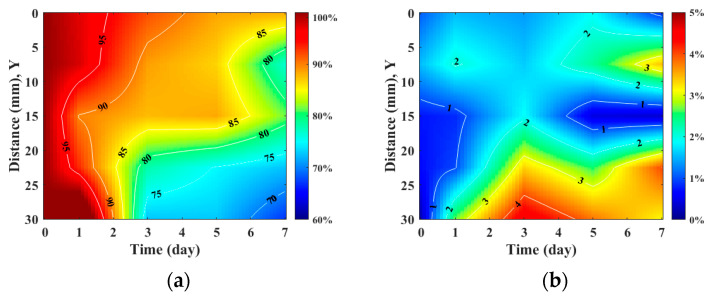
Spatiotemporal mapping of (**a**) relative skin thickness and (**b**) RSD corresponding to different regions of skin flap as a function of post-surgery time. (*N* = 6).

**Figure 9 sensors-24-00363-f009:**
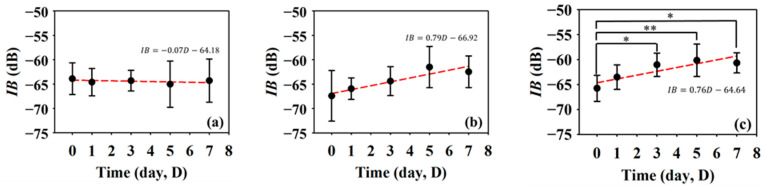
Average *IB*s in the (**a**) proximal, (**b**) middle, and (**c**) distal parts as a function of post-surgery time after. (*N* = 6, *: *p* < 0.05, **: *p* < 0.005).

**Figure 10 sensors-24-00363-f010:**
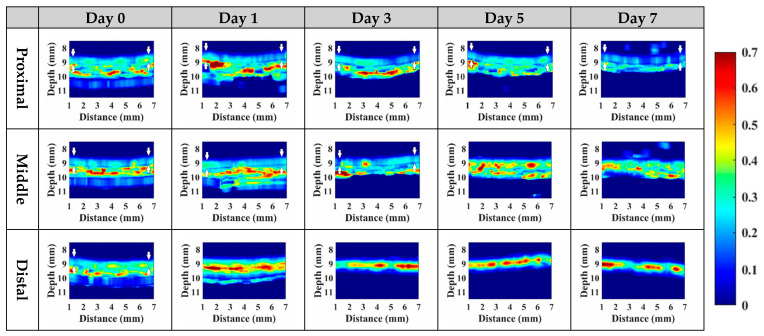
Typical 2D WMC Nakagami images of the proximal, middle, and distal parts of post-surgery skin flap on day 0, 1, 3, 5, and 7.

**Figure 11 sensors-24-00363-f011:**
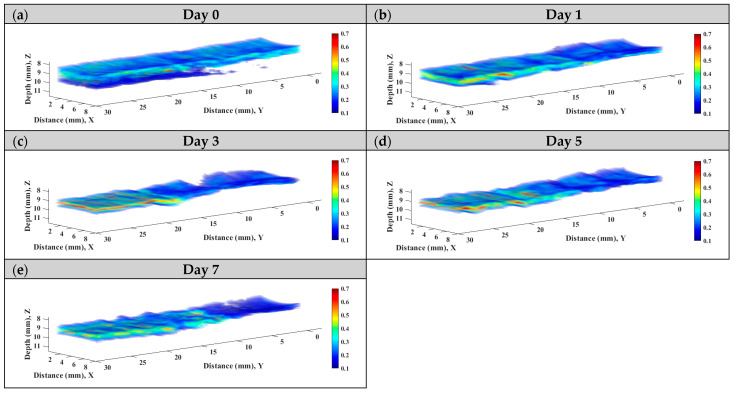
Typical 3D WMC Nakagami images of the skin flap on day (**a**) 0, (**b**) 1, (**c**) 3, (**d**) 5, and (**e**) 7 post-surgery.

**Figure 12 sensors-24-00363-f012:**
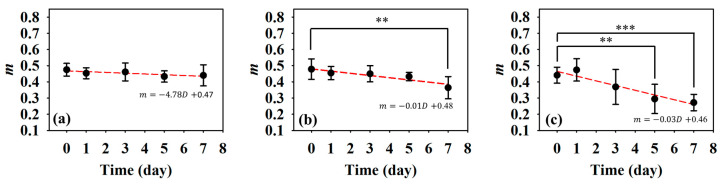
Average Nakagami *m* parameters of the (**a**) proximal, (**b**) middle, and (**c**) distal parts as a function of post-surgery time. (*N* = 6, **: *p* < 0.005, ***: *p* < 0.001).

**Figure 13 sensors-24-00363-f013:**
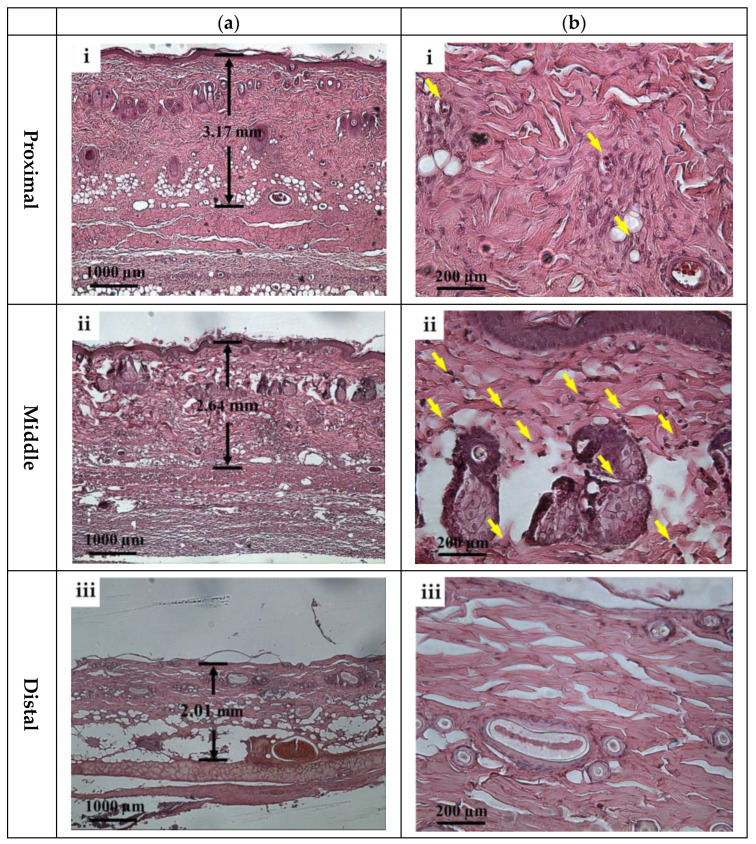
Typical H&E stained histological sections of (**a**) 40× and (**b**) 200× images acquired from (**i**) proximal, (**ii**) middle, and (**iii**) distal parts of tissues of post-surgery skin flap on day 7. Note: Yellow arrows correspond to immune cells.

**Table 1 sensors-24-00363-t001:** Characteristics of the applied transducer.

Center Frequency	26.9 MHz
−6 dB bandwidth	6.5 MHz
*f*-number	1.37
Depth of focus	8.9 mm
Aperture size	6.5 mm

## Data Availability

The data presented in this study are available on request from the corresponding author.

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
