# Peer review of "In Situ Monitoring and Assessment of Ischemic Skin Flap by High-Frequency Ultrasound and Quantitative Parameters"

_sensors, 2024, doi:10.3390/s24020363_

Round 1
Reviewer 1 Report
Comments and Suggestions for Authors
The authors have presented a study about assessment of Ischemic Skin Flap by using High- Frequency Ultrasound and Quantitative Parameters. However there are few concerns that needs to be addressed:
1.The most common skin imaging technologies that are used are skin photography, dermoscope, reflectance confocal microscopy, multispectral optoacoustic tomography, optical coherence tomography, and HFUS. But the authors preferred to use HFUS. So, the authors need to explain its advantages over the other technologies in the introduction section with relevant application examples.
2. The authors have considered QUS parameters where they did not consider the shear wave elasticity, Why? The authors need to justify.
3. The authors need to provide individual figure description to (i), (ii), (iii) and (iv) as they are presented in figures 5. Along with that the authors need to double check all the figures are provided with proper description.
4. Since the authors have presented the block diagram, it would be more appropriate if they can provide the real time view of their apparatus used for their experiment methods and procedure.
5. It is advised to cite most recent works from the last five years. As it is observed that the cited works are very old except the one paper from year 2023.
6. It is advised to provide an abbreviation table at the end of the manuscript for all the short words used in the manuscript.
Comments on the Quality of English Language
English is good
Reviewer 2 Report
Comments and Suggestions for Authors
Please see attached docx file

Please see attached docx file
Round 2
Reviewer 1 Report
Comments and Suggestions for Authors
From the revised manuscript, it is observed that the authors have addressed all the concerns that are highlighted earlier.
Comments on the Quality of English LanguageEnglish is good.